# DocLayout-YOLO: Enhancing Document Layout Analysis through Diverse Synthetic Data and Global-to-Local Adaptive Perception

## Abstract

Document Layout Analysis is crucial for real-world document understanding systems, but it encounters a challenging trade-off between speed and accuracy: multimodal methods leveraging both text and visual features achieve higher accuracy but suffer from significant latency, whereas unimodal methods relying solely on visual features offer faster processing speeds at the expense of accuracy. To address this dilemma, we introduce DocLayout-YOLO, a novel approach that enhances accuracy while maintaining speed advantages through document-specific optimizations in both pre-training and model design. For robust document pre-training, we introduce the Mesh-candidate BestFit algorithm, which frames document synthesis as a two-dimensional bin packing problem, generating the large-scale, diverse DocSynth-300K dataset. Pre-training on the resulting DocSynth-300K dataset significantly improves fine-tuning performance across various document types. In terms of model optimization, we propose a Global-to-Local Controllable Receptive Module that is capable of better handling multi-scale variations of document elements. Furthermore, to validate performance across different document types, we introduce a complex and challenging benchmark named DocStructBench. Extensive experiments on downstream datasets demonstrate that DocLayout-YOLO excels in both speed and accuracy. Code, data, and models will be made publicly available.

## 1 Introduction

With the rapid advancement of large language models and retrieval-augmented generation (RAG) research Lewis et al. (2020); Ram et al. (2023); Edge et al. (2024), the demand for high-quality document content parsing Wang et al. (2024b) has become increasingly critical. A central step in document parsing is Document Layout Analysis (DLA), which aims to precisely locate different types of regions (text, titles, tables, graphics, etc.) within a document. Over the past few years, DLA algorithms have made significant progress, performing well on common document types. However, when faced with diverse document formats, existing layout analysis algorithms Huang et al. (2022); Li et al. (2022) still struggle with speed and accuracy.

Currently, there are two main approaches to document parsing: multimodal methods that combine visual and textual information, and unimodal methods that rely solely on visual features. Multimodal methods, which typically involve pretraining on document images using unified text-image encoders, generally achieve higher accuracy but are often slower due to the complexity of their architectures. In contrast, unimodal methods, which rely only on visual features, offer faster processing speeds but tend to lack accuracy due to the absence of specialized pretraining and model design for document data. To achieve robust performance on diverse real-world documents while meeting the demands of real-time applications, this paper introduces the DocLayout-YOLO layout detection algorithm. This method leverages the strengths of both multimodal and unimodal approaches to quickly and accurately identify various regions within documents. As illustrated in Figure 1, DocLayout-YOLO matches the speed of the mainstream unimodal method YOLOv10 (Wang et al., 2024a) and surpasses all existing methods, including the unimodal DINO-4scale (Zhang et al., 2023a) and YOLO-v10, as well as the multimodal LayoutLMv3 (Huang et al., 2022) and DiT-Cascade (Li et al., 2022), in terms of accuracy on diverse evaluation datasets. Specifically, we

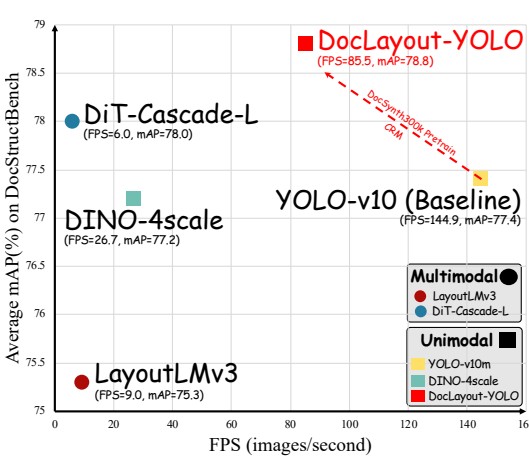

(a) Comparison of speed (FPS) and accuracy (mAP) of DocLayout-YOLO (Ours) against existing methods on the DocStructBench dataset (including Academic, Textbook, Market, and Financial documents).

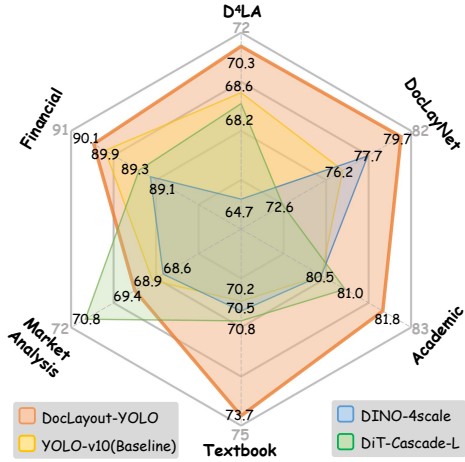

(b) Detailed mAP values of DocLayout-YOLO (Ours) and other methods on $D^4LA$, DocLayoutNet, and the four subsets of the DocStructBench dataset.

Figure 1: Comparisons between DocLayout-YOLO and existing state-of-the-art (SOTA) DLA methods. DocLayout-YOLO surpasses unimodal and multimodal methods in both speed and accuracy.

optimize the YOLOv10 algorithm along two dimensions: pretraining on diverse document data with visual annotations and refining the target detection network structure for document layout analysis.

We observe that multimodal layout analysis methods such as LayoutLMv3 and DiT-Cascade significantly enhance model generalization by pretraining on large-scale unsupervised document data. However, for unimodal layout analysis methods, existing datasets predominantly comprise single document types such as PubLayNet (Zhong et al., 2019) and DocBank (Li et al., 2020b). Models fine-tuned on such datasets tend to overfit to a single distribution, failing to generalize to the diverse layouts encountered in real-world scenarios. To address this, we propose an automated pipeline for constructing diverse document layout data, introducing the Mesh-candidate BestFit algorithm. This algorithm synthesizes document layouts by leveraging principles from the two-dimensional bin packing problem, using a rich set of base components (text, images, tables) to generate a large-scale, diverse pretraining corpus, DocSynth-300K.

YOLO (Jocher et al., 2023; Wang et al., 2024a), a leading algorithm in object detection, excels in both accuracy and speed on natural images. To further enhance YOLO's performance on document images, we adapt the network to the specific characteristics of document data. In diverse documents, the scale of different elements can vary significantly, from small single-line titles to full-page paragraphs, images, and tables. To better handle these multi-scale variations, we introduce the Global-to-Local Controllable Receptive Module (GL-CRM), enabling the model to effectively detect targets of varying scales. The contributions of this paper can be summarized as follows:

- This paper proposes DocLayout-YOLO, a novel model for diverse layout analysis tasks, which leverages the large-scale and diverse document layout dataset DocSynth-300K, and incorporates the GL-CRM to enhance detection performance.
- This paper introduces the Mesh-candidate BestFit algorithm, which synthesizes diverse layout documents from various components (text, images, tables) to create the DocSynth-300K dataset. This dataset will be open-sourced to support further research in document layout analysis.
- This work designs the GL-CRM, which enhances the model's capability to detect elements of varying scales, thereby improving detection accuracy.
- Extensive experiments are conducted on the $D^4LA$, DocLayNet, and our in-house diverse evaluation datasets (DocStructBench). The proposed DocLayout-YOLO model achieves state-of-the-art mAP scores of 70.3%, 79.7%, and 78.8% respectively, along with an inference speed of 85.5 frames per second (FPS), thus enabling real-time layout analysis on diverse documents.

## 2 RELATED WORK

### 2.1 DOCUMENT LAYOUT ANALYSIS APPROACHES

Document Layout Analysis (DLA) focuses on identifying and locating different components within documents, like text and images. DLA approaches are divided into unimodal and multimodal methods. Unimodal methods treat DLA as a special object detection problem, using generic off-the-shelf detectors (Ren et al., 2015; Zhong et al., 2019; Carion et al., 2020; Jocher et al., 2023; Zhang et al., 2023a). Multimodal methods improve DLA by aligning text-visual features through pre-training. For example, LayoutLM (Xu et al., 2020; 2021; Huang et al., 2022) offers a unified approach with various pre-training goals, achieving impressive results on various document tasks. DiT (Li et al., 2022) boosts performance via self-supervised pre-training on extensive document datasets. VGT (Da et al., 2023) introduces grid-based textual encoding for extracting text features.

### 2.2 DOCUMENT LAYOUT ANALYSIS DATASETS

Current document layout analysis datasets, such as the IIT-CDIP (Lewis et al., 2006) with 42 million low-resolution, unannotated images, and its subset RVL-CDIP (Harley et al., 2015), which categorizes 400,000 images into 16 classes, suffer from limitations in annotation detail. PubLayNet (Zhong et al., 2019) includes 360,000 pages from PubMed journals, significantly scaling up the dataset size for document layout analysis. DocBank (Li et al., 2020b) annotates 500,000 arXiv pages using weak supervision, while DocLayNet (Pfitzmann et al., 2022) focuses on 80,863 pages from magazine-type documents. $D^4LA$ (Da et al., 2023) manually annotates 11,092 images from RVL-CDIP across 27 categories, and $M^6Doc$ (Cheng et al., 2023) offers a diverse collection of 9,080 images annotated with 74 types but is not open source due to copyright restrictions. Additional datasets such as DEES200 (Yang et al., 2017), CHN (Li et al., 2020a), Prima-LAD (Antonacopoulos et al., 2009), and ADOPD (Gu et al., 2024) are either not open-sourced or primarily suitable for fine-tuning. As for document generation methods (Zhang et al., 2023b; Inoue et al., 2023; Hui et al., 2023; Jiang et al., 2023; Kong et al., 2022; Gupta et al., 2021), most approaches focuses on academic papers. Overall, current document layout analysis datasets have significant limitations in diversity, volume, and annotation granularity, leading to sub-optimal pre-training models.

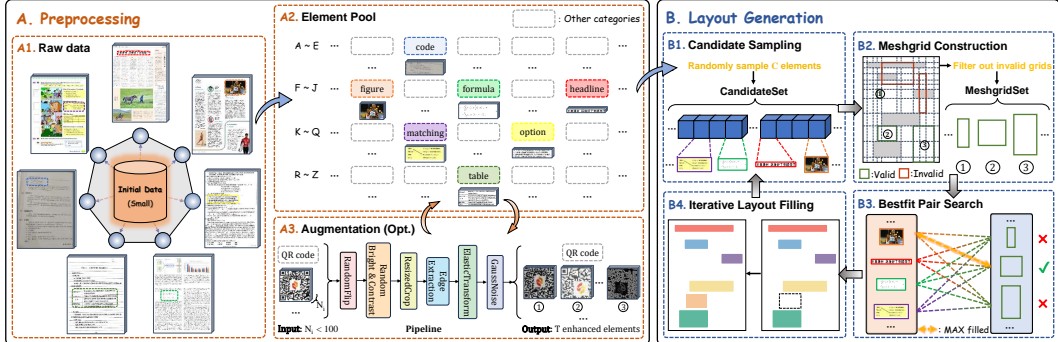

Figure 2: Illustration of Mesh-candidate BestFit. Initially, in *(A) Preprocessing*, a category-wise element pool is created from a small initial dataset. During *(B) Layout Generation*, Mesh-candidate BestFit iteratively searches for the optimal candidate-grid match.

## 3 DIVERSE DOCSYNTH-300K DATASET CONSTRUCTION

Existing unimodal pre-training datasets are characterized by significant homogeneity, primarily comprising academic papers. This limitation substantially hinders the generalization capabilities of pre-trained models. To enhance adaptability to diverse downstream document types, it is imperative to develop a more varied pre-training document dataset.

The diversity of pre-training data can be primarily manifested in two dimensions: (1) Element diversity: This includes a variety of document elements, such as text in different font sizes, tables in various forms, and more. (2) Layout diversity: This encompasses various document layouts, including single-column, double-column, multi-column, and formats specific to academic papers, magazines, and newspapers. In this paper, we propose a novel methodology termed **Mesh-candidate BestFit**, which automatically synthesizes diverse and well-organized documents by leveraging both element

and layout diversity. The resulting dataset, termed **DocSynth-300K**, significantly enhances model performance across various real-world document types. The overall pipeline of Mesh-candidate BestFit is illustrated in Figure 2 and detailed as follows:

## 3.1 PREPROCESSING: ENSURING ELEMENT DIVERSITY

In the preprocessing phase, to ensure the inclusion of a diverse range of document elements, we utilize M$^6$Doc test (Cheng et al., 2023), which consists of 74 different document elements coming from about 2800 diverse document pages, as our initial data. Consequently, we fragment the pages, extracting and constructing an element pool by each fine-grain category. Meanwhile, to maintain diversity within elements of the same category, we design an augmentation pipeline that enlarges the data pool of rare categories that have quantities less than 100 elements (Appendix A.2.2).

## 3.2 LAYOUT GENERATION: ENSURING LAYOUT DIVERSITY

In addressing the challenge of synthesizing diverse layouts, the most straightforward approach is random arrangement. However, random arrangement yields disorganized and confusing layouts, which severely hampers the improvement on real-world documents. Regarding the layout generation models based on Diffusion (Chen et al., 2024; Inoue et al., 2023) or GAN (Jiang et al., 2023; Gupta et al., 2021), existing methods are limited to producing homogeneous layouts such as academic papers, which is insufficient to cover various real-world document layouts.

To ensure layout diversity and consistency with real-world documents, inspired by the 2D bin-packing problem, we regard available grids built by the current layout as "bins" of different sizes and iteratively perform the best matching to generate more diverse and reasonable document layouts, balancing both the layout diversity (randomness) and aesthetics (such as fill rate and alignment). Detailed steps of layout generation are demonstrated as follows:

1. *Candidate Sampling* For each blank page, a subset is obtained through stratified sampling from the element pool based on element size, serving as candidate set. Then, randomly sample an element from the candidate set and place it at a certain position on the page.
2. *Meshgrid Construction* Construct the meshgrid based on the layout and filter out the invalid grids that overlaps with inserted elements. Only the remaining grids will be able to participate in matching with the candidate in subsequent steps.
3. *BestFit Pair Search* For each candidate, traverse all grids that meet the size requirement and search for the Mesh-candidate pair with the maximum fill rate. Subsequently, remove the optimal candidate from the candidate set and update the layout.
4. *Iterative Layout Filling* Repeat step 2 ∼ 3 until no valid Mesh-candidate satisfy the size requirement. Ultimately, random central scaling will be applied to all filled elements separately.

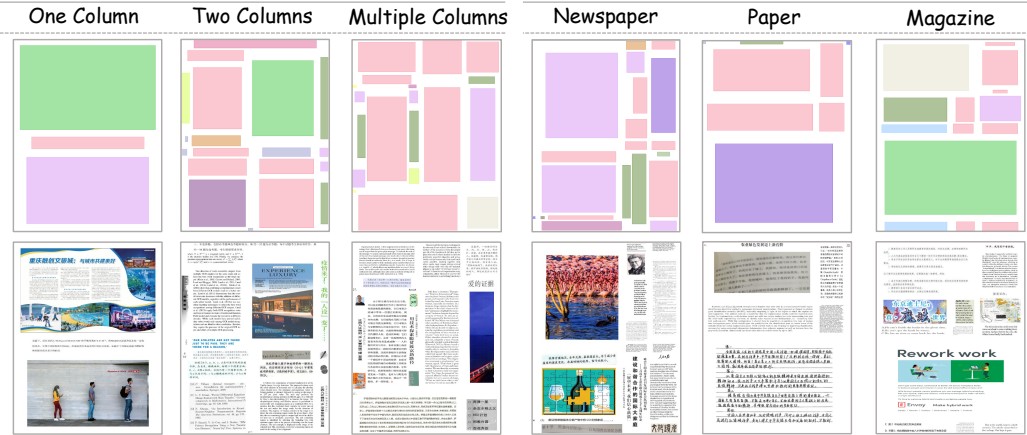

Figure 3: Examples of synthetic document data. Synthetic documents demonstrate comprehensive layout diversity (multiple layout formats) and element diversity (incorporating varied elements).

Through the above process, elements are continuously filled in at optimal positions, ultimately creating a well-organized and visually appealing document image, as shown in Figure 3. The generated documents exhibit a high degree of diversity, which enables the pre-trained models to adapt to a variety of real-world document types effectively. Meanwhile, quantitative analysis demonstrates that the generated document closely adheres to human design principles such as alignment and density (Appendix A.3.1). The detailed algorithm of the above layout generation is shown in Algorithm 1.

## 4 GLOBAL-TO-LOCAL MODEL ARCHITECTURE

Unlike natural images, different elements in document images can vary significantly in scale, such as one-line title and whole-page table. To handle this scale-varying challenge, we introduce a hierarchical architecture called GL-CRM, which consists of two main components: the Controllable Receptive Module (CRM) and the Global-to-Local Design (GL). CRM flexibly extracts and integrates features with multiple scales and granularities, while GL architecture features a hierarchical perception process from global context (whole-page scale), to sub-block areas (medium-scale), and finally local semantics information.

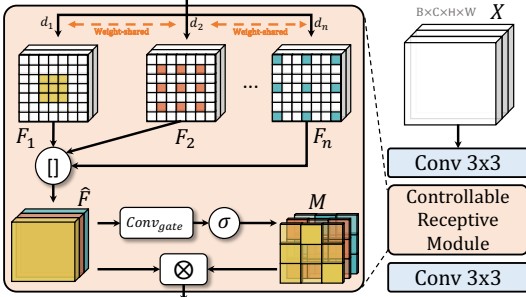

Figure 4: Illustration of Controllable Receptive Module (CRM), which extracts and fuses features of varying scales and granularities.

### 4.1 CONTROLLABLE RECEPTIVE MODULE

CRM is illustrated in Figure 5. To elaborate, for each layer's feature $X$, we start by extracting features using a weight-shared convolution layer $w$ with kernel size $k$. To capture features of different granularities, we employ a set of varying dilation rates $d = [d_1, d_2, ..., d_n]$. This approach allows us to obtain a set of features of different granularities, denoted as $F = [F_1, F_2, \ldots, F_n]$:

$$F_i = GELU(BN(Conv(X, w, d_i)))$$ (1)

After extracting features $F = [F_1, F_2, \ldots, F_n]$ of different granularities, we proceed to integrate these features and allow the network to learn to fuse different feature components autonomously:

$$\hat{F} = Concat([F_1, F_2, \ldots, F_n])$$ (2)

$$M = \sigma(GELU(BN(Conv_{gate}(\hat{F}))))$$ (3)

A lightweight convolutional layer $Conv_{gate}$ with a kernel size of 1 and groups of $nC$ is used to extract a mask $M$ with values ranging between 0 and 1. $M$ can be considered importance weights for different features. Finally, $M$ is applied to the fused features $\hat{F}$, followed by a lightweight output projector $Conv_{out}$. Additionally, a shortcut connection is used to merge the integrated feature with the initial feature $X$:

$$X_{CRM} = X + GELU(BN(Conv_{out}(M \otimes \hat{F})))$$ (4)

The CRM is plugged into the conventional CSP bottleneck (Wang et al., 2020) for extracting and enhancing features of different granularities, as shown in Figure 5. The functionality of the CRM is controlled by two parameters $k$ and $d$, which control the granularity and scale of extracted features.

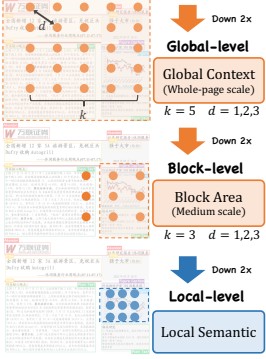

Figure 5: Illustration of Global-to-local design.

### 4.2 GLOBAL-TO-LOCAL DESIGN

***Global-level.*** For the shallow stage, which contains rich texture details, we use CRM with enlarged kernel size and dilation rates ($k = 5$, $d = 1, 2, 3$). A large kernel helps capture more texture details and preserve local patterns for whole-page elements.

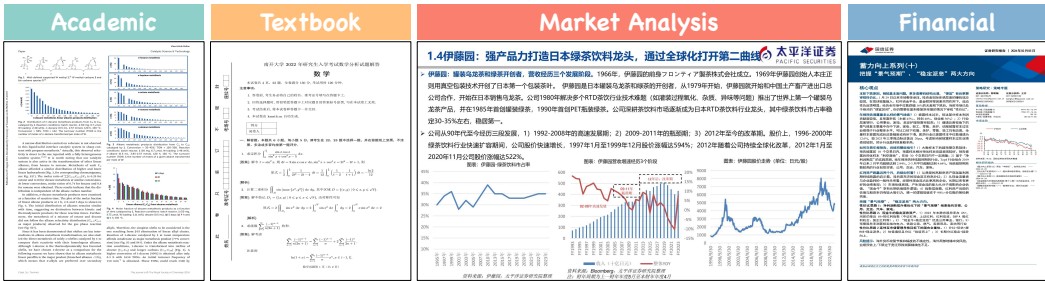

Figure 6: Examples of complex documents with different formats and structures in DocStructBench.

***Block-level.*** For the intermediate stage, where the feature map is downsampled and texture feature is reduced, we employ CRM with smaller kernel ($k = 3, d = 1, 2, 3$). In this case, expanded dilation rates are sufficient for the perception of medium-scale elements, such as document sub-blocks.

***Local-level.*** For the deep stage, where semantic information is predominant, we use a basic bottleneck that serves as a lightweight module which focuses on local semantic information.

## 5 EXPERIMENTS

### 5.1 EXPERIMENTAL METRICS AND DATASETS

For evaluation metrics, we report COCO-style mAP (Lin et al., 2014) for accuracy and FPS (processed images per second) for speed. For evaluation datasets, experiments are conducted on the two most complex public DLA datasets $D^4LA$ (Da et al., 2023) and DocLayNet (Pfitzmann et al., 2022). $D^4LA$ consists of 11,092 noisy images annotated with 27 categories from IIT-CDIP (Lewis et al., 2006) across different 12 document types. The training set consists of 8,868 images and the testing set consists of 2,224 images. As for DocLayNet, DocLayNet contains 80,863 pages from 7 document types and is manually annotated with 11 categories. Images are split into 69,103/6,480/4,994 for training/validation/testing, respectively. DocLayNet validation set is used for evaluation.

Meanwhile, to quantitatively evaluate model performance across different document types, we curate an in-house dataset termed **DocStructBench**, which is a comprehensive dataset designed for evaluation across various real-world scenario documents. It consists of four subsets categorized by the source of the documents: Academic, Textbooks, Market Analysis, and Financial (examples of these documents are illustrated in Figure 6). The data sources of DocStructBench are notably diverse, encompassing a broad range of domains from various institutions, publishers, and websites. DocStructBench consists of 7,310 training images and 2,645 testing images. Each image has been manually annotated across 10 distinct categories: Title, Plain Text, Abandoned Text, Figure, Figure Caption, Table, Table Caption, Table Footnote, Isolated Formula, and Formula Caption. For experiments on DocStructBench, we perform training on a mixture of all four subsets and report results on each subset separately. Other details about DocStructBench can be found at Appendix A.1.

### 5.2 COMPARISON DLA METHODS & DATASETS

DocLayout-YOLO is compared with both multimodal and unimodal methods. Multimodal methods include LayoutLMv3 (Huang et al., 2022), DiT-Cascade (Li et al., 2022), VGT (Da et al., 2023). For unimodal comparison methods we use robust object detector DINO-4scale-R50 (Zhang et al., 2023a). For DLA pre-training datasets, we compare DocSynth-300K with public DLA pre-training datasets PubLayNet (Zhong et al., 2019) and DocBank (Li et al., 2020b).

### 5.3 IMPLEMENTATION DETAILS

For DocLayout-YOLO, we conduct pre-training on DocSynth-300K with image longer side resized at 1600 and use a batch size of 128 and learning rate of 0.02 for 30 epochs. For fine-tuning on DocLayNet, longer side is resized to 1120 and learning rate is set to 0.02. For fine-tuning on $D^4LA$, the longer side is set to 1600 and learning rate is set to 0.04. For fine-tuning on DocStructBench, the

Table 1: Results of DocLayout-YOLO with different optimization strategies. *Pretrain* denotes DocSynth-300K pre-training. Resulting DocLayout-YOLO significantly outperforms the baseline model. $\uparrow \Delta$ denotes improvements compared with baseline YOLO-v10 model.

| Improvement | | D$^4$LA | | DocLayNet | | Academic | | Textbook | | Market Analysis | | Financial | |
|---|---|---|---|---|---|---|---|---|---|---|---|---|---|
| GL-CRM | Pretrain | mAP | AP50 | mAP | AP50 | mAP | AP50 | mAP | AP50 | mAP | AP50 | mAP | AP50 |
| | | 68.6 | 80.7 | 76.7 | 93.4 | 80.5 | 95.0 | 70.2 | 88.0 | 68.9 | 79.2 | 89.8 | 95.9 |
| ✓ | | 69.8 | 81.7 | 77.7 | 93.0 | 81.4 | 95.4 | 71.5 | 88.8 | 70.2 | 80.0 | 90.0 | 95.8 |
| | ✓ | 69.8 | 82.1 | 79.3 | 93.6 | 82.1 | 95.8 | 71.5 | 88.5 | 69.3 | 79.6 | 90.3 | 95.5 |
| ✓ | ✓ | 70.3 | 82.4 | 79.7 | 93.4 | 81.8 | 95.8 | 73.7 | 90.3 | 69.4 | 79.4 | 90.1 | 95.9 |
| $\uparrow \Delta$ | | 1.7 | 1.7 | 3.0 | - | 1.3 | 0.8 | 3.5 | 2.3 | 0.5 | 0.2 | 0.3 | - |

Table 2: Performance comparison on D$^4$LA and DocLayNet. v10m++ denotes the original v10m bottleneck enhanced by our proposed GL-CRM bottleneck. Best and second best are highlighted.

| Methods | | Backbone | Pretrain Dataset | D$^4$LA | | DocLayNet | |
|---|---|---|---|---|---|---|---|
| | | | | mAP | AP50 | mAP | AP50 |
| *Unimodal* | YOLO-v10 | v10m | - | 68.6 | 80.7 | 76.2 | 93.0 |
| | DINO-4scale | R50 | ImageNet1K | 64.7 | 76.9 | 77.7 | 93.5 |
| *Multimodal* | VGT | ViT-B | IIT-CDIP, 42M | 68.8 | - | - | - |
| | LayoutLMv3-B | ViT-B | IIT-CDIP, 42M | 60.0 | 72.6 | 75.4 | 92.1 |
| | DiT-Cascade-B | ViT-B | IIT-CDIP, 42M | 67.7 | 79.8 | 73.2 | 87.6 |
| | DiT-Cascade-L | ViT-L | IIT-CDIP, 42M | 68.2 | 80.1 | 72.6 | 84.9 |
| *Ours* | DocLayout-YOLO | v10m++ | DocSynth, 300K | 70.3 | 82.4 | 79.7 | 93.4 |

longer side is set to 1280 and learning rate is set to 0.04. Training performs with a patience of 100 epochs on 8×A100 GPUs. As for comparison models, DINO employs MMDetection (Chen et al., 2019), using a multi-scale training with an image longer side of 1280 and an AdamW optimizer at $1.0 \times 10^{-4}$. LayoutLMv3 and DiT use Detectron2 Cascade R-CNN (Wu et al., 2019) training with an image longer side of 1333, SGD optimizer of $2.0 \times 10^{-4}$ for 60k iterations.

## 5.4 MAIN RESULTS

### 5.4.1 EFFECTIVENESS OF PROPOSED OPTIMIZATION STRATEGIES

We start by analyzing the effects of different improvement strategies implemented in DocLayout-YOLO, with the experimental results presented in Table 1. Results indicate that *(1) DocSynth-300K largely enhances performance across various document types*, DocSynth-300K pre-trained model achieves 1.2 and 2.6 improvement on D$^4$LA and DocLayNet, which encompasses multiple document types. Meanwhile, DocSynth-300K pre-trained model also leads to improvement on four subsets of DocStructBench. *(2) The resulting DocLayout-YOLO achieves significant improvement*, by combining both CRM and DocSynth-300K pre-training, the resulting DocLayout-YOLO achieves 1.7/2.6/1.3/3.5/0.5/0.3 improvements compared with baseline YOLO-v10 model.

### 5.4.2 COMPARISON WITH CURRENT DLA METHODS

Next, we conduct the comparison with existing DLA methods across multiple datasets. Results of D$^4$LA and DocLayNet are shown in Table 2. We can conclude that *(1) DocLayout-YOLO outperforms robust unimodal DLA methods.* For instance, it shows an improvement of 2.0 over DINO, which is the second best on DocLayNet. *(2) DocLayout-YOLO also outperforms SOTA multimodal methods*. For example, on the D$^4$LA dataset, DocLayout-YOLO achieves 70.3 mAP, surpassing second-best VGT's 68.8. Meanwhile, we conduct experiments on DocStructBench and results are presented in Table 3. DocLayout-YOLO achieves superior performance in three out of four subsets, surpassing existing SOTA unimodal (DINO) and multimodal approaches (DIT-Cascade-L). As for Market Analysis, DocLayout-YOLO is second best compared to DIT-Cascade-L, we suspect this is because DocSynth-300K pre-training is still not sufficient for most complex layouts.

Table 3: Performance comparison on DocStructBench. v10m++ denotes original v10m bottleneck enhanced by our proposed GL-CRM bottleneck. FPS is tested on the same single A100 GPU machine. LayoutLMv3-B[C] denotes pre-trained on additional Chinese document data. * denotes FPS tested under Detectron2, † denotes FPS tested under Ultralytics (Jocher et al., 2023) and ‡ denotes tested under MMDetection. Best and second best are highlighted.

| Method | | Backbone | Academic | | Textbook | | Market Analysis | | Financial | | FPS |
|---|---|---|---|---|---|---|---|---|---|---|---|
| | | | mAP | AP50 | mAP | AP50 | mAP | AP50 | mAP | AP50 | |
| Unimodal | YOLO-v10 | v10m | 80.5 | 95.0 | 70.2 | 88.0 | 68.9 | 79.2 | 89.9 | 95.9 | 144.9† |
| | DINO-4scale | R50 | 80.5 | 95.4 | 70.5 | 85.6 | 68.6 | 79.2 | 89.1 | 95.6 | 26.7‡ |
| Multimodal | DiT-Cascade-B | ViT-B | 79.7 | 95.1 | 69.7 | 86.1 | 63.7 | 71.0 | 88.7 | 94.1 | 14.1* |
| | DiT-Cascade-L | ViT-L | 81.0 | 96.0 | 70.8 | 86.8 | 70.8 | 80.8 | 89.3 | 94.5 | 6.0* |
| | LayoutLMv3-B | ViT-B | 76.5 | 94.9 | 66.0 | 82.3 | 65.7 | 75.2 | 85.7 | 90.4 | 9.0* |
| | LayoutLMv3-B[C] | ViT-B | 77.7 | 93.5 | 68.0 | 82.8 | 67.9 | 75.7 | 87.6 | 92.1 | 9.0* |
| Ours | DocLayout-YOLO | v10m++ | 81.8 | 95.8 | 73.7 | 90.3 | 69.4 | 79.4 | 90.1 | 95.9 | 85.5† |

Table 4: Donwstream fine-tuning performance of different document dataset pre-trained model (baseline YOLO-v10m is utilized). *baseline* row indicates from scratch training results. Results show that compared with public and synthetic document datasets, DocSynth-300K shows better adaptability across all document types. Best and second best are highlighted.

| Data Type | Pretrain Dataset | Volume | Academic | | Textbook | | Market Analysis | | Financial | |
|---|---|---|---|---|---|---|---|---|---|---|
| | | | mAP | AP50 | mAP | AP50 | mAP | AP50 | mAP | AP50 |
| | *baseline* | | 80.5 | 95.0 | 70.2 | 88.0 | 68.9 | 79.2 | 89.9 | 95.9 |
| Public | M⁶Doc | 2k | 80.4 | 94.9 | 70.0 | 87.7 | 68.9 | 79.1 | 89.7 | 95.8 |
| | DocBank | 400k | 81.6 | 95.5 | 70.9 | 89.6 | 69.1 | 79.5 | 90.1 | 95.9 |
| | PubLayNet | 300k | 81.0 | 95.3 | 71.5 | 88.8 | 69.1 | 78.8 | 89.7 | 95.7 |
| Synthetic | Random | 300k | 80.5 | 95.1 | 71.2 | 88.8 | 68.1 | 78.6 | 89.6 | 95.7 |
| | Diffusion | 300k | 80.7 | 95.2 | 71.9 | 89.4 | 68.9 | 79.3 | 89.3 | 95.8 |
| | DocSynth | 300k | 82.1 | 95.8 | 71.5 | 88.5 | 69.3 | 79.6 | 90.3 | 95.5 |

As for inference speed, we carefully evaluate the FPS of various DLA methods, and results show that *(3) DocLayout-YOLO is significantly more efficient than current DLA methods*. Although there is a slight decrease compared to the baseline YOLO-v10, DocLayout-YOLO still demonstrates an obvious advantage in speed. For example, compared with best multimodal methods DIT-Cascade-L, DocLayout-YOLO achieves $14.3\times$ faster FPS. For the best unimodal method DINO, DocLayout-YOLO also shows $3.2\times$ faster FPS.

## 5.5 ABLATION STUDIES

### 5.5.1 COMPARISONS WITH DIFFERENT DOCUMENT SYNTHETIC METHODS

Table 5: Data used in LACE.

| Data | Type | Volume |
|---|---|---|
| DSSE200 | Academic | 271 |
| CHN | Wikipedia | 10K |
| DocBank | Academic | 400K |
| PubLayNet | Academic | 300K |
| DocLayNet | Multiple | 80K |
| D⁴LA | Multiple | 9K |
| Prima-LAD | Multiple | 478 |

In this section, we compare DocSynth-300K with different document synthetic methods to evaluate the quality of synthetic document data. Specifically, we generate documents using different methods while keeping the rendering elements consistent with DocSynth-300K. Consequently, the performance of pre-trained models is evaluated on downstream fine-tuning datasets. The comparative layout generation methods include two approaches: ***Random*** and ***Diffusion***. Random involves arbitrarily arranging the document layouts, whereas, for Diffusion, we train SOTA diffusion-based layout generation method LACE (Chen et al., 2024) using 1M document images from seven downstream datasets to generate layouts (training data used shown in Table 5). Results are conducted on the baseline YOLO-v10 model and the experimental results are shown in Table 4.

From results, we can conclude that: *(1) Random layouts is unsuitable for document pre-training.* Though certain improvements are observed, the performance of random layout is suboptimal due to large misalignments with real documents. *(2) Diffusion layout is limited to certain document*

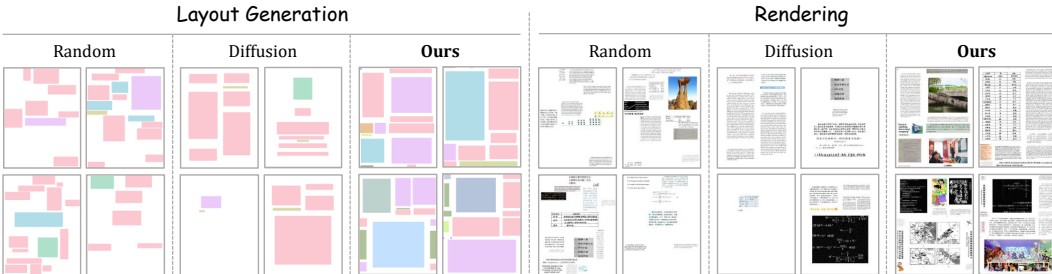

Figure 7: Visualization of generated document images using different document synthetic methods.

***types.*** Models pre-trained with Diffusion layouts outperform Random, likely because Diffusion produces layouts that more closely resemble actual documents. However, these layouts exhibited limited diversity, leading to improvement on limited types such as Academic and Textbook. ***(3) DocSynth-300K shows superior generalization ability across various document types.*** Compared to both Diffusion and Random, model pre-trained with DocSynth-300K leads to improvement on all four subsets and shows superior generalization ability. Both quantitative and visualization results (as shown in Figure 7) demonstrate that our proposed synthetic pipeline can generate documents with much greater diversity and higher quality.

### 5.5.2 COMPARISONS WITH PUBLIC DOCUMENT PRE-TRAINING DATASETS

Comparison results with public document pre-training datasets are shown in Table 4. It can be concluded that ***DocSynth-300K features a more effective document pre-training dataset compared with public datasets.*** Firstly, for $M^6$Doc test dataset, where the elements of DocSynth-300K come from, suffers from severe overfitting due to its limited size. Secondly, for PubLayNet and DocBank, although they feature large volumes of data, the limited element diversity (less than 10 element categories) and layout diversity (only academic paper) lead to a less diversified feature representation in the pre-trained models, which constrain further improvement (though certain improvements are observed) and fail to consistently enhance generalization ability on all downstream datasets. In contrast, for DocSynth-300K, the pre-trained model achieves comprehensive improvements and outperforms PubLayNet and DocBank on most downstream datasets, demonstrating that DocSynth-300K is much more effective for improvement on various downstream documents.

### 5.5.3 ABLATIONS ON EFFECTS OF GL-CRM

Table 6: Ablation studies on GL-CRM.

| Ablation | | D⁴LA | | | | |
|---|---|---|---|---|---|---|
| Global-level | Block-level | mAP | AP50 | $AP_s$ | $AP_m$ | $AP_l$ |
| | | 68.6 | 80.7 | 47.0 | 53.2 | 68.8 |
| ✓ | | 69.2 ↑0.6 | 81.2 ↑0.5 | 47.1 ↑0.1 | 53.9 ↑0.7 | 69.6 ↑0.8 |
| | ✓ | 69.3 ↑0.7 | 81.5 ↑0.8 | 47.2 ↑0.2 | 55.0 ↑1.8 | 69.4 ↑0.6 |
| ✓ | ✓ | 69.8 ↑1.2 | 81.7 ↑1.0 | 47.2 ↑0.2 | 55.3 ↑2.1 | 70.2 ↑1.4 |

Finally, we conduct ablation study on the proposed GL-CRM, with the results shown in Table 6. The experiments demonstrate that the inclusion of the Global level significantly enhances detection accuracy for medium and large objects. Furthermore, incorporating the Block-level results in the most substantial improvement for medium objects, corresponding to sub-blocks existing in documents. Experiments validate the effectiveness of global to local design of GL-CRM.

## 6 CONCLUSION

In this paper, we propose DocLayout-YOLO, which excels in both speed and accuracy. DocLayout-YOLO incorporates improvements from both pre-training and model optimization perspectives: For pre-training, we propose the Mesh-candidate BestFit methodology, which synthesizes a high-quality, diverse DLA pretraining dataset, DocSynth-300K. For model optimization, we introduce the GL-CRM, enhancing the network's perception of document images from a hierarchical global-block-local manner. Experimental results on extensive downstream datasets demonstrate that DocLayout-YOLO significantly outperforms existing DLA methods in both speed and accuracy.

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

# A APPENDIX

In the appendix, we provide detailed information on our proposed in-house evaluation dataset Doc-StructBench (Appendix A.1), as well as details of Mesh-candidate BestFit and more visualization examples of generated documents (Appendix A.2). Next, we give a quantitative evaluation of DocSynth-300K data from a design principle perspective (Appendix A.3.1), as well as ablation studies on pre-training data volume (Appendix A.3.2). Finally, the detection examples of DocLayout-YOLO on multiple kinds of real-world documents are demonstrated (Appendix A.4).

## A.1 DOCSTRUCTBENCH DETAILS

Table 7: Document source and train/test split of Docstructbench.

| Type | Source | Training | Testing |
|---|---|---|---|
| Academic | Academic papers | 1605 | 402 |
| Textbook | Textbooks & Test papers | 2345 | 587 |
| Analysis Report | Industry & market analysis report | 2660 | 651 |
| Financial | Financial business document | 2472 | 592 |

Docstructbench is a diverse and complex document structure dataset comprising 9,082 training images and 2,232 test images. It includes four subsets: Academic, Textbook, Market Analysis, and Financial. The distribution and sources of documents in each subset are detailed in Table 7. The instances of each document component category are detailed in Table 8.

Table 8: Fine-grained category and number of instances annotated in Docstructbench.

| Category | Interpretation | Training | Testing |
|---|---|---|---|
| Title | Includes multi-level headings, separate lines, bolded, and in a distinct font from the text. | 11384 | 2943 |
| Plain text | Main body text of the document. | 45243 | 12455 |
| Abandon | Includes headers, footers, page numbers, page footnotes, and marginal notes. | 16640 | 4379 |
| Figure | Isolate figure floating in the document. | 5164 | 1296 |
| Figure caption | Corresponding caption interpreting the figure. | 2660 | 715 |
| Table | Isolate table floating in the document. | 1389 | 407 |
| Table caption | Corresponding caption interpreting the table. | 911 | 271 |
| Table footnote | The footnote of a table, typically provides additional explanations and clarifications about the table. | 1490 | 370 |
| Isolate formula | A standalone equation (excluding equations embedded within the text) | 795 | 221 |
| Formula caption | The caption of a formula, typically refers to the label or numbering of the formula. | 385 | 86 |

## A.2 MESH-CANDIDATE BESTFIT

### A.2.1 ALGORITHM OF LAYOUT GENERATION

The algorithm of layout generation is detailed as Algorithm 1, which iteratively searches for the best matches between the candidate and all grids (bins). After the best matching pair is found, the candidate is inserted into the document and continues to iteratively search for the optimal match until the number of elements reaches a threshold $N$ (empirically set to 15). The matching threshold $fr_{thr}$ is set to $10^{-4}$.

### A.2.2 DATA AUGMENTATION PIPELINE

In the preprocessing phase, we conduct a specifically designed augmentation pipeline for rare categories that have few elements in the element pool. The details are as follows:

1. **Random Flip** Considering the various possibilities of text orientation in different documents, we enhance the original data with random flips in both the horizontal and vertical directions at a probability of 0.5.
2. **Random Brightness & Contrast** We simulate the real-world environments under a wide variety of lighting conditions and brightness levels by randomly altering the brightness and contrast of elements at a probability of 0.5.
3. **Random Cropping** To guide the model to concentrate more on local features, we employ a probability of 0.7 to perform random cropping on the elements within the area range of $0.5 \sim$ 0.9.
4. **Edge Extraction** We use the Sobel filter to perform edge detection and extract the contour information within the elements with a probability of 0.2, thereby enhancing the richness of the features.

**Algorithm 1:** Mesh-candidate BestFit Algorithm

**Input:** Element pool $P$, $C_{set} = \{e_1, e_2, ..., e_N\}$ sampled from $P$, matching threshold $fr_{thr}$;
**Output:** Generated layout $L$;

1 sample $e^*$ from $C_{set}$ and insert into $L$;
2 **while** $|L| < N$ **do**
3      $M_g$ = MeshEngine($L$);
4      **foreach** $candidate\ e_i \in C_{set}$ **do**
5          **foreach** $meshgrid\ g_j \in M_{set}$ **do**
6              $fr$ = match($e_i, g_j$);
7              **if** $fr > fr_{max}$ **then**
8                  $fr_{max} \leftarrow fr, C_{best} \leftarrow e_i, M_{best} \leftarrow g_j$;
9              **end**
10          **end**
11      **end**
12      **if** $fr_{max} < fr_{thr}$ **then**
13          **break**
14      **else**
15          remove $C_{best}$ from $C_{set}$ and insert $C_{best}$ into $L$;
16      **end**
17 **end**
18 **return** $L$;

5. ***Elastic Transformation & Gaussian Noisification*** We distort and blur the original data through a slight elastic transformation and a Gaussian noise addition process to simulate jitter or resolution-induced distortion in reality.

### A.2.3 OTHER DETAILS

In the layout generation phase, we iteratively perform the best matching to search for the candidate-grid pair with the highest fill rate until no valid pair satisfies the size requirement. Furthermore, we add an additional restriction, namely that the number of small elements must not exceed $Mini_{num}$, since a surplus of small elements leads to a layout that does not adhere to conventional aesthetic standards. Specifically, $Mini_{num}$ is set to 5.

### A.2.4 MORE VISUALIZATION EXAMPLES

Here, a richer visualization of the generated data is shown in Figure 8. S, M, L respectively denote small, medium, and large elements, indicating the components that are relatively abundant on the page. It is evident that the data we generate is rich in categories and possesses strong diversity. It can not only generate dense layouts containing many small elements but also produce sparse layouts composed of a few large elements, similar to the layouts generated by diffusion-based models.

### A.3 MORE EVALUATION EXPERIMENTS

### A.3.1 EVALUATION OF SYNTHETIC DOCUMENT FROM DESIGN PRINCIPLE PERSPECTIVE

In this section, we quantitatively evaluate whether the synthetic document data aligns with the human design principle. The evaluation employs the *Align* and *Density* metrics, which respectively measure the aesthetic quality of layouts in terms of document alignment and density. For *Align*, we utilize the LayoutGAN++ (Kikuchi et al., 2021; Li et al., 2021) metric which measures the alignment of elements in the document:

$$L_{alg} = \sum_{i=1}^{N} \min \left( \begin{array}{c} g(\Delta x_i^L), g(\Delta x_i^C), g(\Delta x_i^R) \\ g(\Delta y_i^T), g(\Delta y_i^C), g(\Delta y_i^B) \end{array} \right). \tag{5}$$

where $x_i^*(* = L, C, R), y_i^*(* = T, C, B)$ denotes the x-axis left/center/right and y-axis top/center/bottom of $i$-th elements in document, $g(x) = -\log(1-x)$, and $\Delta x_i^*(* = L, C, R)$

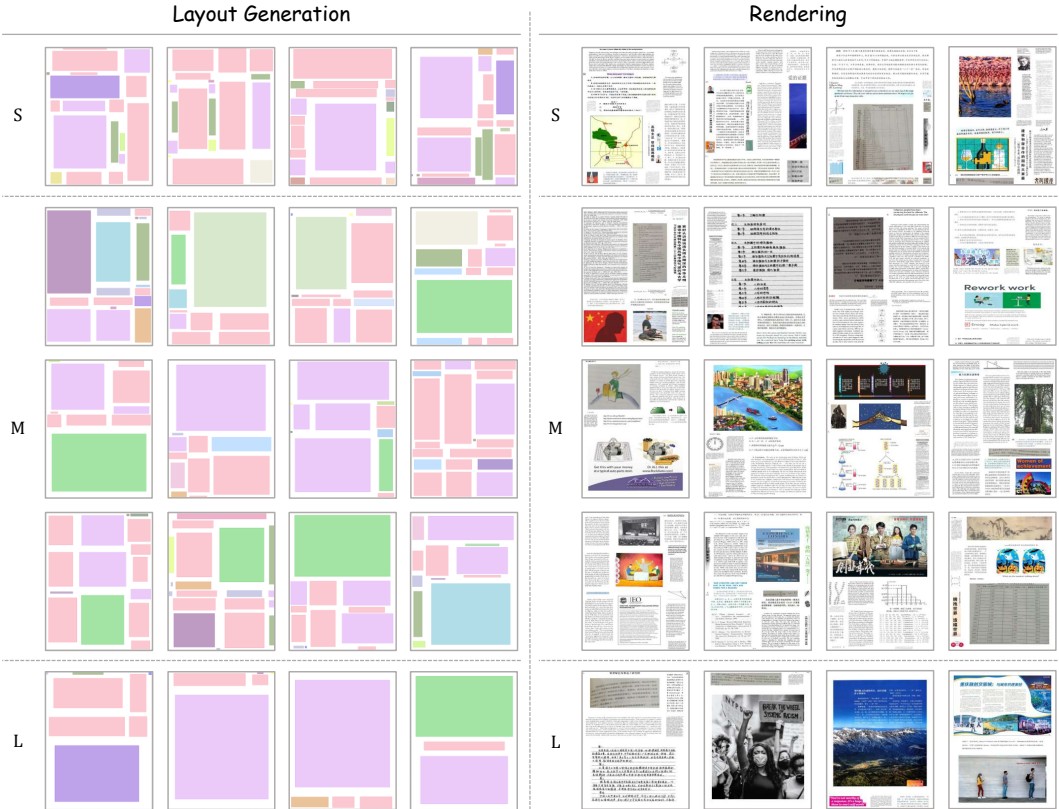

Figure 8: Visualization of generated diverse layouts and corresponding pages after rendering. *S, M, L* respectively denote small, medium, and large elements, indicating the components that are relatively abundant on the page.

is computed as:

$$\Delta x_i^* = \min_{\forall j \neq i} |x_i^* - x_j^*| \tag{6}$$

$\Delta y_i^*(* = T, C, B)$ can be computed similarly. For *Density*, we calculate the ratio of filled area in the layout:

$$L_{dst} = \frac{\sum_{i=1}^N |e_i|}{|L|} \tag{7}$$

Table 9: Quantative comparison between different layout generation methods.

| Layout Generation | Align↓ | Density↑ |
|---|---|---|
| Random | 0.0171 | 0.259 |
| Diffusion (LACE) | 0.0032 | 0.476 |
| **Mesh-candidate BestFit (ours)** | **0.0009** | **0.645** |

where $|e_i|$ denotes area of element $e_i$ in $L$, and $|L|$ denotes area of the whole layout. For *Align*, a lower value denotes a more aligned document. For *Density*, a larger value denotes a more compact and dense layout. The experimental results, as shown in Table 9, indicate that the Mesh-candidate BestFit method significantly outperforms diffusion and random methods in both alignment and density. Visual results further confirm that the layouts produced by Mesh-candidate BestFit better conform to the standards of human aesthetics and design.

### A.3.2 ABLATIONS ON PRETRAINING DATA VOLUME

We conduct ablation experiments on pre-training data volume. We pretrain basic YOLO-v10 using 0-500K Mesh-candidate BestFit generated pre-training data and fine-tune on D⁴LA dataset subsequently. Results are shown in Figure 9. In our experiments, we observe a distinct correlation betw-

een pre-training data volume and model performance. Specifically, for data less than 100k, there is a consistent improvement in model performance correlating with an increase in data volume. However, model performance shows noticeable fluctuations when the data volume reaches 200k. Notably, model performance reaches its top when the data volume increases to 300k.

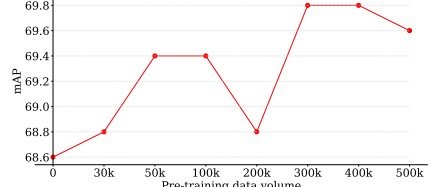

Figure 9: Ablations on pre-training data volume.

## A.4 DETECTION EXAMPLES

In Figures 10 and Figure 11, we demonstrate the detection examples of DocLayout-YOLO after fine-tuning with the DocStructBench dataset on various types of downstream documents. Examples show that the model, fine-tuned using the DocStructBench dataset, effectively adapts to multiple document types, showcasing considerable practicality and coverage.

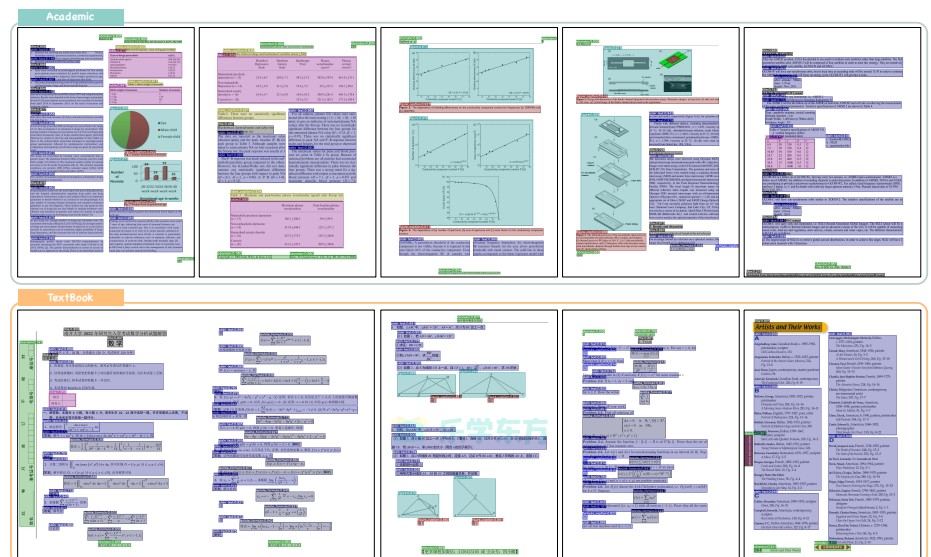

Figure 10: Detection results of DocLayout-YOLO on Academic and Textbook subsets.

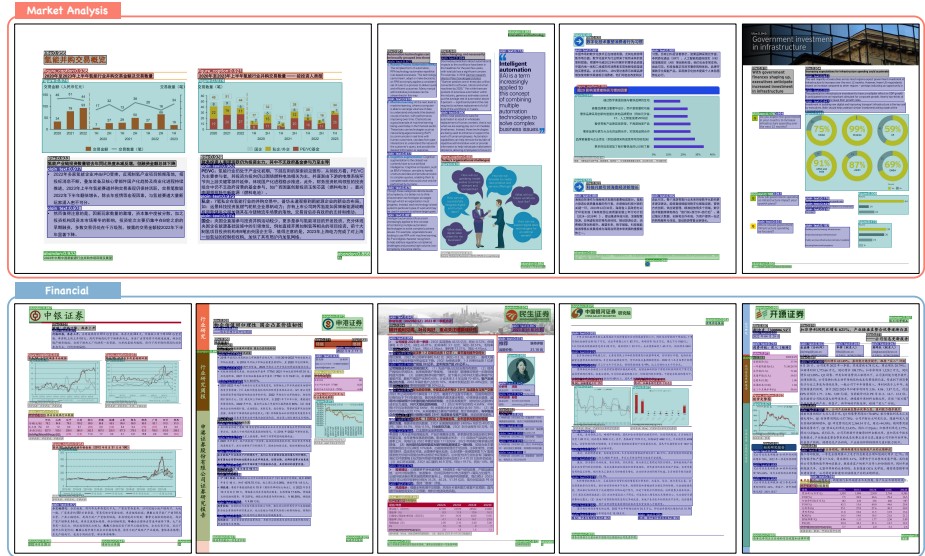

Figure 11: Detection results of DocLayout-YOLO on Market Analysis and Financial subsets.

