# OpenReview forum: "DocLayout-YOLO: Enhancing Document Layout Analysis through Diverse Synthetic Data and Global-to-Local Adaptive Perception"
_ICLR.cc/2025/Conference — ICLR 2025 Conference Withdrawn Submission_

### Official Review · Reviewer_n77e · 2024-10-23

**Soundness:** 2
**Presentation:** 2
**Contribution:** 2
**Rating:** 3
**Confidence:** 5

**Summary:**

This paper proposes DocLayout-YOLO for diverse layout analysis tasks. In addition, this paper also proposes Mesh condition BestFit for synthesizing layout analysis datasets, and proposes GL-CRM to further improve the accuracy of the model.

**Strengths:**

1, The proposed GLOBAL-TO-LOCAL method can improve the accuracy of Yolo in layout analysis tasks.
2, The proposed methodology termed Mesh-candidate BestFit can automatically synthesize diverse documents.

**Weaknesses:**

But I think this paper has significant flaws:
1. The author's experimental results are not convincing. The author claims that their results achieved SOTA, but did not compare with better models in the DocLayNet dataset, such as VGT (Da, Cheng, et al. "Vision grid transformer for document layout analysis." ICCV 2023.), GLAM  (Wang, Jilin, et al. "A graphical approach to document layout analysis." ICDAR 2023.), etc. These models have achieved better accuracy on DocLayNet.
2. In addition, the author did not compare the results with other experiments on larger benchmarks (DocBank, PublayNet). These datasets are authoritative benchmarks, similar to COCO in general object detection. However, the author's results on the in-house dataset DocStructBench lack authority.
3. The dataset DocSynth-300K synthesized by the author is not large enough, and some publicly available datasets such as DocBank possesses 500K data. Since the author has proposed a method for synthesizing datasets, the amount of data should not be less than some open-source datasets, and may even be more than 3 times larger than the largest dataset. Because the pre-trained dataset needs to be much larger than the fine-tuned dataset to demonstrate the effectiveness.
4. The innovation of this paper is limited, and the proposed model is based on Yolo without significant structural innovation.

**Questions:**

Why doesn't the author synthesize more pre-training data to demonstrate effectiveness?

---

### Official Review · Reviewer_hXwc · 2024-10-31

**Soundness:** 3
**Presentation:** 3
**Contribution:** 2
**Rating:** 6
**Confidence:** 4

**Summary:**

This work DocLayout-YOLO proposes a new technique of document layout analysis. First, they have created a large-scale dataset DocSynth-300K for robust document-specific pre-training where they synthesize documents as a 2D bin packing problem. They solve it with the Mesh-candidate BestFit algorithm where they treat available grids built by the input layout as bins of different sizes and iteratively perform the best matching to generate more diverse and reasonable document layouts. Also, they improve the existing YOLO-v10 through a global-to-local controllable receptive module for better optimization.

**Strengths:**

This paper has the following two contributions:
1. A large-scale diverse document pertaining dataset with Mesh-candidate best-fit algorithm.
2. Model optimization with Global-to-local Controllable Receptive Module.

**Weaknesses:**

1. The main concern is the motivation of the paper. In document layout analysis the problem isn't getting better performance for the well-defined class with structured layout. In contrast, the goal is to improve the performance of small-scale diverse datasets like the following:
PRIMA: https://www.primaresearch.org/datasets/Layout_Analysis
Historical Japanese dataset: https://dell-research-harvard.github.io/HJDataset/
There are no particular contributions regarding the performance improvement on these small-scale diverse datasets.

2. In the related works, Some Unimodal works have not been cited, which are pretty effective and faster compared to the proposed one.
SwinDocSegmenter: https://arxiv.org/abs/2305.04609
SemiDocSeg: https://link.springer.com/article/10.1007/s10032-024-00473-y
GraphKD: https://arxiv.org/abs/2402.11401

3. The main drawback of Meshgrid construction is the layout will be always structured (i.e. it ensures all the document elements must have a non-overlapping rectangular region). How they tackle complex layout which has overlapping region and unstructured documents like magazines and comics.

4. How does the GL-CRM impact computational efficiency, especially for larger models or when scaling? As it is just a combination of GeLU activation over batch-normalization. Why do other activation fails? Does LayerNorm can also affect performance?

**Questions:**

1. Motivation for creating a large-scale structured document pretraining dataset?
2. Why the recently proposed vision-based unimodal works has been ignored?
3. How does the model tackle overlap and unstructured documents?
4. Effectiveness of GL-CRM during scaling and with larger models

**Details Of Ethics Concerns:**

There are no ethical concerns related to this paper.

---

### Official Review · Reviewer_x4z7 · 2024-11-02

**Soundness:** 3
**Presentation:** 1
**Contribution:** 3
**Rating:** 5
**Confidence:** 3

**Summary:**

The paper presents two contributions. On one hand, it devises the Mesh-candidate BestFit algorithm to create the DocSynth-300K dataset for enhanced document pre-training. On the other hand, it proposes a Global-to-Local Controllable Receptive Module for model optimization.

**Strengths:**

This paper makes two main contributions. First, it introduces the Mesh-candidate BestFit algorithm for robust document pre-training, generating the large-scale DocSynth-300K dataset which improves fine-tuning performance across various document types. Second, it proposes DocLayout-YOLO with a Global-to-Local Controllable Receptive Module for model optimization, enabling better handling of multi-scale variations in document elements and achieving excellent performance in both speed and accuracy.

**Weaknesses:**

The author's contributions in terms of both data and model design seem acceptable. As I am not in this field, I still need to refer to the opinions of other reviewers and the chair. I have some concerns as follows.

1. When the author introduces the two types of methods in the introduction, references should be added. Furthermore, you can introduce how some classic methods are done.

2. The author is extremely reluctant to add references throughout the paper, which is very unfriendly to researchers who are not in this field. In addition, references should also be added in the tables. Otherwise, it is not clear whether these are the author's own reproductions or results from other papers.

3. The performance comparisons in Table 2, Table 3, and Table 4 are unfair as the pre-training data of each method is different.

4. In various other fields, the feature fusion of global-local seems to be a common practice and there is nothing new about it.

5. The author seems not to have mentioned Figure 4 in the manuscript.

**Questions:**

See the weaknesses.

---

### Official Review · Reviewer_v6jU · 2024-11-05

**Soundness:** 2
**Presentation:** 2
**Contribution:** 3
**Rating:** 5
**Confidence:** 4

**Summary:**

The paper introduces the Mesh-candidate BestFit algorithm to generate the large-scale DocSynth-300K dataset, which improves fine-tuning results across diverse document types. It also proposes a Global-to-Local Controllable Receptive Module to effectively handle multi-scale variations in document elements. Experimental results show that DocLayout-YOLO delivers good performance in both speed and accuracy on  $D^4LA$, DocLayNet and in-house datasets DocStructBench.

**Strengths:**

-The paper introduces a new synthesized dataset, DocSynTH-300K, based on M6Doc, offering an approach for generating diverse layouts.
-Experiments conducted on in-house data demonstrate the effectiveness of the proposed method.
-The figures and charts are visually appealing.

**Weaknesses:**

- The authors state in line 153 on page 3 that "unimodal pre-training datasets are characterized by significant homogeneity, primarily comprising academic papers." This statement is inaccurate. In fact, there are already datasets from diverse domains, such as M6Doc, which is a large-scale, multi-domain dataset. Moreover, the authors’ own DocSynth-300k dataset is constructed based on M6Doc, further illustrating the availability of varied data sources.
- The description of layout generation is not very clear. How does the candidate sampling and meshgrid construction process ensure that elements are placed effectively on a page layout without overlapping?
- The experiments conducted on in-house data are not very convincing. It would be better to include a different backbone model pre-trained on DocSynth-300k and test it on public datasets.
- Lack of comparison with SOTA: "M2Doc: A Multi-Modal Fusion Approach for Document Layout Analysis,AAAI 2024"

**Questions:**

- The small initial dataset is come from $M^6Doc$ test, why not choose $M^6doc$ training part as initial dataset then perform test on $M^6Doc$ test?
- What role does stratified sampling play in selecting elements from the pool?

---

### Note · Authors · 2024-11-14

**Comment:**

Thanks for reviews' valuable suggstions. I withdraw this paper in behalf of paper authors.

**Withdrawal Confirmation:**

I have read and agree with the venue's withdrawal policy on behalf of myself and my co-authors.